# Mitigating Impact of Spatial Variance of Turbulence in Wind Energy Applications

Jonas Kazda[1] and Jakob Mann[1]

[1]DTU Wind Energy, Frederiksborgvej 399, 4000 Roskilde, Denmark

**Correspondence:** Jakob Mann (jmsq@dtu.dk)

**Abstract.** For the first time an analytical solution for the quantification of the spatial variance of the second-order moment of correlated wind speeds was developed in this work. The spatial variance is defined as random differences in the sample variance of wind speed between different points in space. The approach is successfully verified using simulation and field data. The impact of the spatial variance on three selected applications relevant to the wind energy sector is then investigated including mitigation measures. First, the difference of the second-order moment between front-row wind turbines of Lillgrund wind farm is investigated. The variance of the difference ranges between 25% and 48% for turbulence intensities ranging from 7% to 10% and a sampling period of 10min. It is thus suggested to use the second-order moment measured at each individual turbine as input to flow models of wind farm controllers in order to mitigate random error. Second, the impact of the spatial variance of the measured second-order moment on the verification of wind turbine performance is investigated. Misalignment between the mean wind direction and the line connecting the meteorological mast and wind turbine is observed to result in an additional random error in the observed second-order moment of wind speed. In the investigated conditions the random error was up to 34%. Such random error adds uncertainty to the turbulence intensity-based classification of the fatigue loads and power output of a wind turbine. To mitigate the random error it is suggested to either filter the measured data for low angles of misalignment, or to quantify wind turbine performance using the ensemble averaged measurements of the same wind conditions. Third, the verification of sensors in wind farms was investigated with respect to the impact of distant reference measurements. In case of a misalignment between the wind direction and the line connecting sensor and reference, an increased random error will hamper the comparison of the measured second-order moments. The suggested mitigation measures are equivalent to those for the verification of turbine performance.

## 1 Introduction

The wind energy market has been growing rapidly at a rate of 16% throughout the past decade reaching 539,123MW of global, installed capacity in 2017 (Global Wind Energy Council (GWEC), 2017). Many areas of the wind energy sector require measurements of the wind turbulence in the atmospheric boundary layer (ABL), which is typically quantified as turbulence intensity. Turbulence intensity is defined according to the IEC norm (International Electrotechnical Commission, 2005) as the ratio of the square root of the second-order moment of axial wind speed to the mean axial wind speed of the same 10-min period. Common devices for the measurement of the second-order moment of wind speed are sonic anemometers, cup anemometers

and lidars. These devices provide an estimate of the second-order moment covering a confined volume of the ABL, that is at the sensor location, or in the case of lidars, along the laser beam. However, out of economical and/or technical considerations, these measurements are, at times, performed spatially separated from the desired location. Typically, then extrapolation is used to estimate the second-order moment of wind speed at the desired location. In other applications, measurements from multiple locations are aggregated to obtain statistical measures. However, second-order moments of wind speed measured at different locations over a finite period will be different, even under homogeneous conditions. This is because the coherence of turbulence decreases with distance, particularly the larger the cross-flow separation (Sørensen et al., 2012). To quantify and analyze the spatial variance of the second-order moment of wind speed, an analytical approach is therefore developed. The spatial variance is defined as random differences in the sample variance of wind speed between different points in space.

The spatial variance is relevant to a variety of areas in the wind energy sector. This work focuses on three areas, that is wind farm control, the verification of wind turbine performance, and sensor verification. In wind farm control the operation of wind turbines in a wind farm is coordinated with the objective to either (i) maximize the total power production of the wind farm (Kazda et al., 2016), or (ii) to follow a target level for the total power output of the wind farm, while optionally reducing fatigue loads of wind turbines (Kazda et al., 2018). Advanced wind farm controllers typically employ models of wind farm operation (Gebraad et al., 2016; Kazda and Cutululis, 2018) to predict the impact of the control on the flow within the wind farm. Increasingly, turbulence intensity is used as input to flow models (Niayifar and Porté-Agel, 2016; Göçmen et al., 2018). The most common turbulence-related input is ambient turbulence intensity, which is typically obtained from measurements at upstream turbines of the wind farm. There are two commonly-employed approaches to process these measurements for use in the flow model. In one approach, the turbulence measurements at each upstream turbine define the ambient turbulence intensity at the respective turbine. In the other approach, the ambient turbulence intensity is defined as the average of the turbulence intensity at all upstream turbines. Because of the distance between wind turbines, the measured second-order moment of wind speed however varies between the turbines. The present work thus investigates the impact of the spatial variance on the two above described approaches for defining ambient turbulence intensity.

Next, it is increasingly common to use turbulence intensity measurements for the classification of turbine performance. Having been discussed in literature over the past decade, the impact of turbulence intensity on turbine performance will be addressed in the next revision of the IEC standard (IEC, 2005). This is because the turbulence intensity in the flow that approaches a wind turbine influences its fatigue loads (Eggers et al., 2003; Saranyasoontorn and Manuel, 2008) and power output (Elliott and Cadogan, 1990; Gottschall and Peinke, 2008; Clifton and Wagner, 2014). In the process of verifying wind turbine characteristics, the turbulence intensity in the free flow is typically measured at a meteorological mast adjacent to the wind turbine. As a result of the distance between mast and wind turbine, the spatial variance of the second-order moment can impact the accuracy of the measured turbulence intensity. Uncertainty in the measured turbulence intensity propagates into the uncertainty of the measured power curve and fatigue loads of the wind turbine. When the mast location is directly upstream, the random error due to the spatial variance of turbulence can be regarded as small assuming Taylor's hypothesis of frozen turbulence. The assumption may lead to an underestimation of the difference in variances in the situation where one measurement position is more or less directly downstream from the other. In case of an offset of the mast location orthogonal

to the direction of wind flow, a random error results, because of the spatial variance of turbulence. The magnitude of the impact and approaches for its mitigation are therefore investigated in the present work.

The third application area discussed in the present work is the verification of spatially separated sensors for the measurement of the second-order moment of wind speed. In such case, the measurements of the to-be-verified sensor are compared to a spatially distant reference measurement. Due to the distance between sensor and reference, the result of the verification can be corrupted by the spatial variance of the second-order moment of wind speed. This phenomenon is discussed in the present work on the example of the verification case in Mittelmeier et al. (2016). Here turbine-based measurements of turbulence intensity are verified with reference measurements at an adjacent meteorological mast.

The remainder of this paper is structured as follows. In section 2 the developed analytical solution for the quantification of the spatial variance of the second-order moment of wind speed is detailed. In section 3, first, the analytical solution is verified, and thereafter, the mitigation of the impact of the spatial variance of the second-order moment of wind speed is discussed for three selected applications. The paper is concluded with a summary of the key findings in section 4.

## 2 Analytical Solution to Spatial Variance of Second-order Moment of Wind Speed

In the following the first analytical solution is derived for the quantification of the expected spatial variance of the second-order moment of wind speed measured over a time period $T$ at two spatially-separated points $\boldsymbol{a}$ and $\boldsymbol{b}$. The expected spatial variance of the second-order moment of wind speed $\delta\mu_{2,L,a-b}^2$ in an arbitrary directional component projection $L$ is defined as

$$\delta\mu_{2,L,\boldsymbol{a}-\boldsymbol{b}}^2(T) = \langle[\mu_{2,L,\boldsymbol{a}}(T) - \mu_{2,L,\boldsymbol{b}}(T)]^2\rangle \tag{1}$$

where $\mu_{2,L,\boldsymbol{a}}(T)$ and $\mu_{2,L,\boldsymbol{b}}(T)$ are second-order moments measured at the points $\boldsymbol{a} = (a_x, a_y, a_z)$ and $\boldsymbol{b} = (b_x, b_y, b_z)$ in the ABL. The direction of the component projection $L$ of the measurement is assumed to be the same at the points $\boldsymbol{a}$ and $\boldsymbol{b}$. $x$, $y$, and $z$ are the coordinates of the Cartesian coordinate system. $x$ is set to the mean direction of wind flow, $y$ is the horizontal coordinate orthogonal to $x$, and $z$ the vertical coordinate. The measured second-order moment of wind speed $\mu_{2,L}(T)$ is defined as

$$\mu_{2,L}(T) = \frac{1}{T}\int_{-\frac{T}{2}}^{\frac{T}{2}} (u_L(t) - \overline{u}_L)^2 dt \tag{2}$$

where $u_L(t)$ is the wind speed in an arbitrary component projection $L$ and $\overline{u}_L$ is the wind speed in the same direction averaged over the time interval $[-T/2, T/2]$. The wind speed in an arbitrary component projection $L$ is defined as

$$u_L = \boldsymbol{n}_L \cdot \boldsymbol{u} \tag{3}$$

where $\boldsymbol{n}_L = (n_{L,x}, n_{L,y}, n_{L,z})$ is the unit-directional vector in the direction of the projection, and $\boldsymbol{u} = (u, v, w)$ is the wind velocity.

Assuming a homogeneous turbulent field, the expected spatial variance of the second-order moment (Eq. 1) can be reformulated as

$$5 \quad \delta\mu_{2,L,\boldsymbol{a}-\boldsymbol{b}}^2(T) = 2[\langle\mu_{2,L}(T)^2\rangle - \langle\mu_{2,L,\boldsymbol{a}}(T)\mu_{2,L,\boldsymbol{b}}(T)\rangle] \tag{4}$$

Next, it is assumed that the mean wind speed $\overline{u}_L$ is zero. This assumption is not necessary since assuming a non-zero mean wind speed gives the exact same results, but it makes the equations much more compact. Furthermore, it is assumed that $u_L(t)$ can be represented by a Gaussian process. Consequently, Isserlis' Theorem (Isserlis, 1916, 1918) can be applied to Eq. 4 resulting in

$$10 \quad \delta\mu_{2,L,\boldsymbol{a}-\boldsymbol{b}}^2(T) = \frac{4}{T^2}\left[\iint_{-\frac{T}{2}}^{\frac{T}{2}} \langle u_L(t)u_L(t')\rangle^2 dtdt' - \iint_{-\frac{T}{2}}^{\frac{T}{2}} \langle u_{L,\boldsymbol{a}}(t)u_{L,\boldsymbol{b}}(t')\rangle^2 dtdt'\right] \tag{5}$$

The expected spatial and temporal correlation of wind speeds can be expressed using the two-point correlation tensor of wind velocity $\mathbf{R}(\boldsymbol{r}, \Delta t)$. $\boldsymbol{r}$ is a three-dimensional vector connecting the two points, and $\Delta t$ the time delay. Hence, Eq. 5 can be transformed into

$$\delta\mu_{2,L,\boldsymbol{a}-\boldsymbol{b}}^2(T) = \frac{4}{T^2}\left[\iint_{-\frac{T}{2}}^{\frac{T}{2}} (\boldsymbol{n}_L^T\mathbf{R}(\boldsymbol{0}, t-t')\boldsymbol{n}_L)^2 dtdt' - \iint_{-\frac{T}{2}}^{\frac{T}{2}} (\boldsymbol{n}_L^T\mathbf{R}(\boldsymbol{a}-\boldsymbol{b}, t-t')\boldsymbol{n}_L)^2 dtdt'\right] \tag{6}$$

15      The correlation tensor can be obtained from the infinite volume integral of the spectral tensor $\boldsymbol{\Phi}(\boldsymbol{k})$ as

$$\iint_{-\frac{T}{2}}^{\frac{T}{2}} (\boldsymbol{n}_L^T\mathbf{R}(\boldsymbol{a}-\boldsymbol{b}, t-t')\boldsymbol{n}_L)^2 dtdt' = \iint_{-\frac{T}{2}}^{\frac{T}{2}} \left[\iiint_{-\infty}^{\infty} \boldsymbol{n}_L^T\boldsymbol{\Phi}(\boldsymbol{k})\boldsymbol{n}_L \exp\left(i\boldsymbol{k}(\boldsymbol{a}-\boldsymbol{b} + \begin{pmatrix} U & 0 & 0 \end{pmatrix}(t-t'))\right)dk_1dk_2dk_3\right]^2 dtdt' \tag{7}$$

The spectral tensor $\boldsymbol{\Phi}(\boldsymbol{k})$ can be obtained using the model of Mann (1994). $\boldsymbol{k}$ is the three-dimensional wave number vector. The three dimensional infinite integral over the wavenumber space is denoted as $\int d\boldsymbol{k} = \iiint_{-\infty}^{\infty} dk_1dk_2dk_3$ in the following. The time delay $\Delta t$ is eliminated using Taylor's hypothesis of frozen turbulence as the spatial separation $\Delta x = U(t-t')$ in axial

20      flow direction. $U$ is the mean wind speed in axial flow direction when averaging over the time interval $[-T/2, T/2]$. Expanding above equation and solving the time integral yields

$$\iint\limits_{-\frac{T}{2}}^{\frac{T}{2}} (\boldsymbol{n}_L^T \mathbf{R}(\boldsymbol{a}-\boldsymbol{b},t-t')\boldsymbol{n}_L)^2 dt dt' =$$

$$\int\int (\boldsymbol{n}_L^T \boldsymbol{\Phi}(\boldsymbol{k})\boldsymbol{n}_L)(\boldsymbol{n}_L^T \boldsymbol{\Phi}(\boldsymbol{k}')\boldsymbol{n}_L)\exp\left(i(\boldsymbol{k}+\boldsymbol{k}')(\boldsymbol{a}-\boldsymbol{b})\right)\operatorname{sinc}^2\left(\frac{(k_1+k_1')TU}{2}\right)T^2 d\boldsymbol{k}d\boldsymbol{k}' \quad (8)$$

The derived equation 8 is used in the original problem (Eq. 6) yielding an analytical solution for the spatial variance of the second-order moment

$$\delta\mu_{2,L,\boldsymbol{a}-\boldsymbol{b}}^2(T) = 4\int\int (\boldsymbol{n}_L^T\boldsymbol{\Phi}(\boldsymbol{k})\boldsymbol{n}_L)(\boldsymbol{n}_L^T\boldsymbol{\Phi}(\boldsymbol{k}')\boldsymbol{n}_L)\Big[1-\cos\left((\boldsymbol{k}+\boldsymbol{k}')(\boldsymbol{a}-\boldsymbol{b})\right)\Big]\operatorname{sinc}^2\left(\frac{(k_1+k_1')TU}{2}\right)d\boldsymbol{k}d\boldsymbol{k}' \quad (9)$$

In the following, the normalized spatial variance of the second-order moment $\delta M_{2,L,a-b}$ is defined as

$$\delta M_{2,L,\boldsymbol{a}-\boldsymbol{b}} = \frac{\sqrt{\delta\mu_{2,L,\boldsymbol{a}-\boldsymbol{b}}^2}}{\langle\mu_{2,L}(T)\rangle} \quad (10)$$

The normalization is performed using the ensemble second-order moment of wind speed $\langle\mu_{2,L}(T)\rangle$, which is calculated as

$$\langle\mu_{2,L}(T)\rangle = \iiint\limits_{-\infty}^{\infty} \boldsymbol{n}_L^T\boldsymbol{\Phi}(\boldsymbol{k})\boldsymbol{n}_L\Big[1-\operatorname{sinc}^2\left(\frac{k_1 TU}{2}\right)\Big]d\boldsymbol{k} \quad (11)$$

## 3 Results & Discussion

In the following, the analytical solution is compared to the spatial variance observed in a simulated wind field in order to demonstrate its validity. Thereafter, the mitigation of the impact of the spatial variance is investigated for three selected applications, that is wind farm control, verification of turbine performance and sensor verification.

### 3.1 Verification of Analytical Solution

The analytical solution (Eq. 9) is successfully verified in the following against a simulated wind field.

#### 3.1.1 Simulation Set-up

The turbulent wind field is created using the simulation approach of the Mann model (Mann, 1998). The simulation domain has the dimensions of 5000m × 600m × 600m in the $x$, $y$, and $z$ direction, respectively. The geometric characteristics of the simulation domain and grid are summarized in Table 1.

| Direction | x | y | z |
|---|---|---|---|
| Dimension | 5000m | 600m | 600m |
| Grid points | 1024 | 128 | 128 |
| Grid spacing | 4.88m | 4.69m | 4.69m |

**Table 1.** Key characteristics of domain and grid of simulated wind field.

### 3.1.2 Atmospheric Conditions

The ABL is characterized by the following conditions in both the simulations and the analytical solution. The stability of the ABL is neutral, and hence, the spectral parameters of the Mann model are set to $\alpha\epsilon^{\frac{2}{3}} = 1$, $l = 50$m, $\Gamma = 3.2$, according to Sathe et al. (2013). The value of $\alpha\epsilon^{\frac{2}{3}}$ is arbitrary and irrelevant, since the spatial variance is only considered in a normalized manner. The mean wind speed in the mean wind direction is 8m/s. The duration of averaging $T$ is set to 10min, as it is used for turbulence measurements in the IEC norm (International Electrotechnical Commission, 2005).

### 3.1.3 Comparison with Simulation

Figure 1 compares the spatial variance of the second-order moment obtained using the analytical solution with the results from the simulations. The comparison was conducted for the second-order moment of axial wind speed $u$ for spatial separation of measurement points $a$ and $b$ in the $y$ and $z$ direction. The spatial variance of the second-order moment was normalized by the expected second-order moment, as described in equation 10. The analytical solution was evaluated using adaptive multidimensional numerical integration (Genz and Malik, 1980; Berntsen et al., 1991). The integration range in the analytical solution was adjusted to the simulation domain and grid spacing with the aim to mimic the conditions of the simulations. As such the integration range of $k_1$ of $\mathbf{k}$ was set to $[-\frac{2\pi}{l_{spacing}}, -\frac{2\pi}{l_{domain}}] \cup [\frac{2\pi}{l_{domain}}, \frac{2\pi}{l_{spacing}}]$. $l_{domain}$ and $l_{spacing}$ are the domain size and grid spacing, respectively. The integration range of the other integrals of the analytical solution remained infinite.

The overall agreement of the results shown in Figure 1 demonstrates the validity of the analytical solution. The agreement is better for close separation distances of up to 50m. Here, the mean absolute difference is only 2.6% and 0.55% for separation in $y$ and $z$ direction, respectively. It can further be observed that the spatial variance is generally larger in the simulation results than in the analytical solution. The difference is the result of the underlying assumptions of the simulations and the numerical integration. The numerical error in the integration of the analytical solution is 1% of the result, and hence considered too small to explain the difference. A possible explanation can be that the adjusted integration range in the analytical solution did not fully capture the effects of the simulation grid. Generally, the simulation grid causes a lower ensemble variance of axial wind speed $\langle\mu_{2,u}\rangle$ in the simulations. A lower value in $\langle\mu_{2,u}\rangle$ results because turbulent eddies smaller than the grid spacing cannot be captured. Since $\langle\mu_{2,u}\rangle$ is used to normalize the spatial variance, a lower value in $\langle\mu_{2,u}\rangle$ results in a larger value of the normalized spatial variance in the simulations.

The same trend of the spatial variance can be observed in the results of the analytical solution and of the simulation. As such, the spatial variance of the second-order moment of wind speed increases with larger distance between the measurement

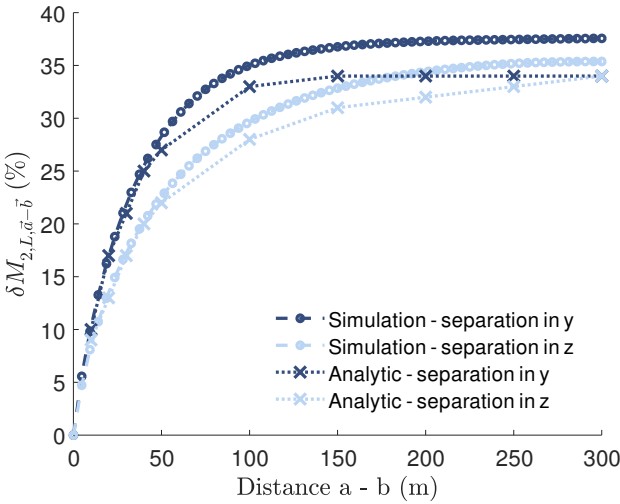

**Figure 1.** Simulation-based validation of analytical calculation of spatial variance of second-order moment of ABL wind flow. Comparison is conducted for second-order moment of axial wind speed $u$ for spatial separation of two points in the cross-axial and vertical direction.

points. This is due to a decreasing coherence of wind turbulence with larger separation distance (Sørensen et al., 2012). At large separation distances, the spatial variance converges to an asymptotic value. The simulation results converge to 0.37 and 0.35 for separation in $y$ and $z$ direction, respectively. The results of the analytical solution converge to 0.34. The asymptotic behaviour can be understood from the analytical solution, particularly from the behaviour of the term $[1 - \cos((\boldsymbol{k} + \boldsymbol{k'})(\boldsymbol{a} - \boldsymbol{b}))]$

in equation (9). For large distances between the measurement points $\boldsymbol{a}$ and $\boldsymbol{b}$, the oscillations of the cosine term are much faster than the change of the remainder of the integrand. As a result, the integral over one period of one minus the cosine is well approximated by the remainder of the integrand. Consequently, the cosine term can be neglected for large separation distances. As a result, the spatial variance of the second order moment converges to an asymptote. Furthermore, when neglecting the cosine term for large separation distances, the integrand becomes independent of the direction of the spatial separation. Hence,

the value of the asymptote is the same for separation in the $y$ direction and the $z$ direction, as observed in the results of the analytical solution in Figure 1 at a distance of 300m.

The asymptotic behaviour of the spatial variance can also be understood from the statistics of uncorrelated variables. At large separation distances, typically in the order of the integral length scale $l$, the second-order moment of wind speed at the points $\boldsymbol{a}$ and $\boldsymbol{b}$ becomes uncorrelated. As a result, the variance of the difference between the second-order moment at the points

$\boldsymbol{a}$ and $\boldsymbol{b}$, that is $\delta\mu_{2,L,\infty}^2$, is twice the variance of the second-order moment of wind speed $\sigma(\mu_{2,L}(T))^2$. Hence, the asymptotic value of the normalized spatial variance can be calculated as

$$\delta M_{2,L,\infty} = \frac{\sqrt{\delta\mu_{2,L,\infty}^2}}{\langle\mu_{2,L}(T)\rangle} \tag{12}$$

$$= \frac{\sqrt{2\sigma^2(\mu_{2,L}(T))}}{\langle\mu_{2,L}(T)\rangle} \tag{13}$$

The variance of the second-order moment of wind speed $\sigma^2(\mu_{2,L}(T))$ can be approximated according to Lenschow et al. (1994). Eq. 12 can thus be approximated using the integral time scale $\tau$ or the integral length scale $\mathcal{L}$ as

$$\frac{\sqrt{2\sigma^2(\mu_{2,L}(T))}}{\langle\mu_{2,L}(T)\rangle} \approx 2\sqrt{\frac{\tau}{T}} \tag{14}$$

$$= 2\sqrt{\frac{\mathcal{L}}{l}} \tag{15}$$

This approximation is valid for $\tau << T$ and $\mathcal{L} << l$. The length scale $l$ is obtained from the measurement duration $T$ and the average axial wind speed $U$ as $l = TU$. For the case shown in Figure 1, the asymptotic value of the normalized spatial variance results as 41%, and hence, is comparable to the results of the analytical solution and of the simulation.

In addition to the effect of the separation distance, Figure 1 shows that the spatial variance of the second-order moment increases faster in the cross-axial direction $y$ than in the vertical direction $z$. This is the result of stronger spectral-coherence of turbulence in the $z$-direction than in the $y$-direction.

### 3.2 Mitigation of Impact in Applications

The spatial variance of the second-order moment of wind speed impacts a variety of applications in the wind energy sector. For the present work, three areas were selected for more detailed discussion, that is wind farm control, the verification of wind turbine performance, and sensor verification.

#### 3.2.1 Wind Farm Control

In wind farm control, recent flow models (Kazda et al., 2018; Göçmen et al., 2018; Niayifar and Porté-Agel, 2016), increasingly use measurements of turbulence intensity as input, particularly ambient turbulence intensity. An approach to obtain an estimate of ambient turbulence intensity is to average turbulence intensity measured at upstream turbines of the wind farm. Such approach, however, introduces a random error into the flow modelling due to the difference of the measured turbulence intensity between the turbines. Using the average, ambient turbulence intensity as input to the flow model results in a deviation of the modelled turbulence intensity from the actual turbulence intensity at a wind turbine. It can be shown using the standalone Dynamic Wake Meandering (sDWM) wind farm operation model (Keck, 2015) that the thereby resulting error in the prediction of power at a downstream turbine can be in the same order of magnitude as the deviation in turbulence intensity. To mitigate this error, the turbulence measured at each turbine location can be used as input to the flow model, as in Göçmen et al. (2016); Kazda et al. (2018). Thereby, the local realization of turbulent flow can be taken into account.

The difference in turbulence intensity between turbines arises from the spatial variance of the second-order moment of wind speed. To demonstrate the magnitude of the spatial variance in a real wind farm, an investigation was made on the westerly, front row of turbines of Lillgrund wind farm. Lillgrund wind farm is located offshore, south-east of Copenhagen, Denmark (Nilsson et al., 2015). A schematic of the westerly front row turbines and the adjacent meteorological mast is shown in Figure 2,

together with the investigated sector of wind direction, wind speed and turbulence intensity. The front row comprises five wind turbines spaced by approximately five rotor diameters, that is 450m. In the following, the spatial variance is first investigated using the analytical solution, and thereafter using measurements. Given a wind direction from west, that is 270°, the spacing of wind turbines orthogonal to the average direction of wind flow is more than 400m. Hence, the turbulence measurements at wind turbines in the front row are separated with at least that distance. For the atmospheric conditions investigated with Figure

1 and the separation distance of more than 400m, the spatial variance is at the asymptotic value of 34% given an averaging time $T$ of 10min.

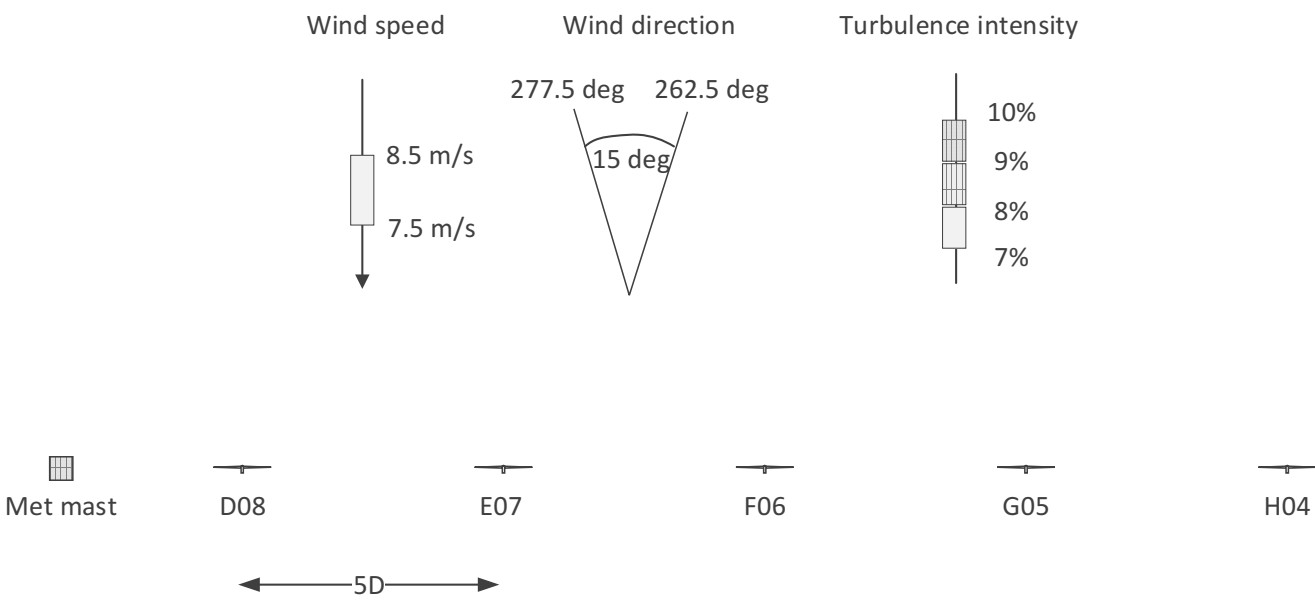

**Figure 2.** Top view of westerly front row of Lillgrund wind farm and investigated range of wind conditions. Variance of second-order moment of nacelle wind speed between wind turbines is investigated with turbine no. D08 for reference.

A similar, spatial variance can also be observed in measurements from Lillgrund wind farm. The spatial variance was investigated using data from more than 7 years of measurements. The ensemble average of the second-order moment of wind speed at the turbines was obtained for ensembles of the following wind condition. The 10min-average wind speed is between

7.5m/s and 8.5m/s, and hence comparable to the wind speed used in the results of the analytic solution above. The 10min-average wind direction is in the 15° westerly sector between 262.5° and 277.5°. The measured 10min-average wind speed and the 10min-based second-order moment of wind speed are obtained from the nacelle anemometry of wind turbines. The

wind direction is measured at the meteorological mast south of the wind turbine row on turbine hub height. The measurements are filtered according to the above described sectors of average wind speed and wind direction. The spatial variance of the second-order moment of axial wind speed $\delta M_{2,u,X-D08}$ is defined with reference to wind turbine D08 as

$$\delta M_{2,u,X-D08} = \frac{\sqrt{\langle(\mu_{2,u,X} - \mu_{2,u,D08})^2\rangle}}{\langle\mu_{2,u,D08}\rangle} \tag{16}$$

where $\mu_{2,u,D08}$ and $\mu_{2,u,X}$ are the second order moments of axial wind speed measured at wind turbine D08 and one of the other front-row wind turbines, respectively.

Figure 3 shows the spatial variance of the second-order moment between the front-row wind turbines E07, F06, G05, H04 and wind turbine D08. The spatial variance is normalized by the ensemble variance of axial wind speed at wind turbine D08. The results are binned with respect to turbulence intensity. The results of each bin are based on at least 33 distinct measurements,

and hence are considered statistically significant.

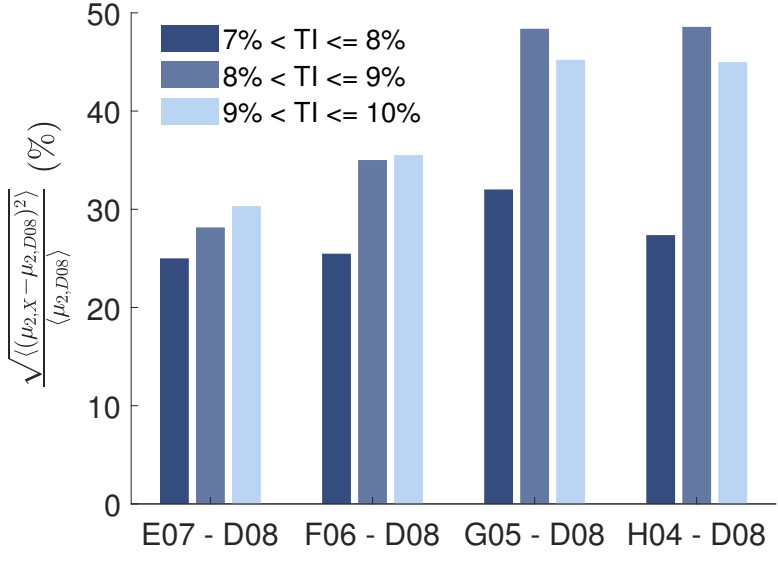

**Figure 3.** Effect of separation distance and atmospheric stability on variance of second-order moment of nacelle wind speed between the western front-row turbines of Lillgrund wind farm. Atmospheric stability is implicitly characterized using turbulence intensity.

Figure 3 shows the effect of separation distance and atmospheric stability on the spatial variance of the second-order moment. Atmospheric stability is implicitly characterized by turbulence intensity. Larger turbulence intensity is likely to correspond to more unstable atmospheric conditions. For turbulence intensities ranging between 7% and 8%, the spatial variance remains similar with larger distance between the turbulence measurement location, except for the spatial variance between turbines

G05 and D08. It is thus likely that the spatial variance has reached the asymptotic value already for the separation distance between turbines E07 and D08. The observed behaviour is thus in line with the result of the analytical solution, which predicts

reaching the asymptotic value in neutral conditions at a separation of 300m. The magnitude of the spatial variance is similar between the measurements and the analytical solution. The spatial variance calculated using the analytical solution is 30% for neutral ABL conditions. The spatial variance observed in the measurements is 25%, except between turbines G05 and D08. The lower spatial variance observed in the field data is due to differences in the power spectrum of wind speed, which mainly occurs

out of two reasons. First, it is likely that the turbulence length scale $L$ of the ABL conditions differ between the measurements and the analytical solution. The ABL conditions in the measurements are implicitly characterized by turbulence intensity, yet cannot be attributed to a specific condition. The ABL conditions used for the analytical solution is neutral. Second, the Mann model, that is used in the analytical solution, provides spectra of undisturbed atmospheric flow. The measurements, however, are made on the nacelle of a wind turbine, where the rotor and nacelle of the wind turbine disturb the flow. The spectrum

of wind speed measured at the nacelle of a wind turbine is expected to contain more energy at higher frequencies than the free flow. In (Crespo and Hernández, 1996) this is shown for the near wake, where the energy in the spectrum of wind speed is increased particularly at higher frequencies. Thus, in the nacelle-based measurements an increased share of the energy of higher frequency eddies is expected in the second-order moment. In case of a larger share of energy at high frequencies the integral length scale $\mathcal{L}$ is smaller. According to Eq. 14 a smaller integral length scale results in a smaller spatial variance, and

thus confirms the reasoning for the observed difference between measurements and analytical solution.

For turbulence intensities ranging between 8% to 10%, the spatial variance converges to an asymptotic value with larger separation between the turbine pairs. The asymptote is reached at the separation distance of 1400m between turbines G05 and D08. It is concluded that the asymptote is reached, since the spatial variance stays constant with the larger separation distance of 1850m between turbines H04 and D08. This asymptotic behaviour observed in the measurements verifies the convergence

of the spatial variance to an asymptotic value, as it is observed in the analytical solution and the simulations. The value of the asymptote is however up to 90% larger than for the results on turbulence intensities ranging between 7% and 8%. The larger value of the asymptote is attributed to more unstable ABL conditions, which is underpinned by the following two observations. First, according to Sathe et al. (2013) the integral length scale is larger in unstable conditions than in stable ones, and based on Eq. 14 this results in a larger value of the asymptote. Second, the higher turbulence intensity of 8% to 10% indicates a more

unstable condition of the ABL. In general, when moving across the stability spectrum from stable to unstable the integral length scale increases (Sathe et al., 2013). As a result of this increase, the asymptotic value of the spatial variance of the second-order moment of wind speed increases, according to Eq. 14.

To conclude, the up to 48% spatial variance observed in the measurements demonstrates that using the average turbulence intensity as input to flow models would result in a considerable random error from the actual turbulence intensity at each

upstream turbine. Hence, it is of advantage to use the locally measured turbulence intensity as input to flow models.

### 3.2.2   Verification of Wind Turbine Performance

The turbulence intensity in the flow approaching a wind turbine influences its fatigue loads (Eggers et al., 2003; Saranyasoon-torn and Manuel, 2008) and power output (Elliott and Cadogan, 1990; Gottschall and Peinke, 2008; Clifton and Wagner, 2014). It is therefore increasingly investigated to classify wind turbine performance, that is, in the present work, power output and

fatigue loads, in terms of turbulence intensity levels. To do so, turbulence intensity is typically obtained from measurements of wind speed upstream of the respective wind turbine. Uncertainty in the upstream measurements of turbulence intensity results in uncertainty in the classification of wind turbine performance.

It is therefore of interest to understand the factors driving the uncertainty in such turbulence intensity measurements and
5    to develop mitigation measures. Turbulence intensity is the ratio of the second-order moment of wind speed and its mean. Due to the distance between measurement location and wind turbine, a source of uncertainty can be the spatial variance of the second-order moment of wind speed. The magnitude of the spatial variance is therefore investigated in the following for a typical set-up used for the verification of wind turbine performance. The results give insight into the impact of the spatial variance on the uncertainty in power output and fatigue loads, and demonstrate the need for mitigation measures.

10    Figure 4 shows a typical experimental set-up used for the verification of the performance of a wind turbine. A meteorological mast adjacent to the wind turbine is used for the measurement of the flow that approaches the wind turbine. In the present study, the distance between the mast and the wind turbine is set to 200m, which is a magnitude comparable to real set-ups. Two cases on the alignment between the mean wind direction and the mast and the wind turbine are shown in the figure. In the case of alignment, the spatial variance of turbulence can be regarded as small assuming Taylor's hypothesis of frozen turbulence.

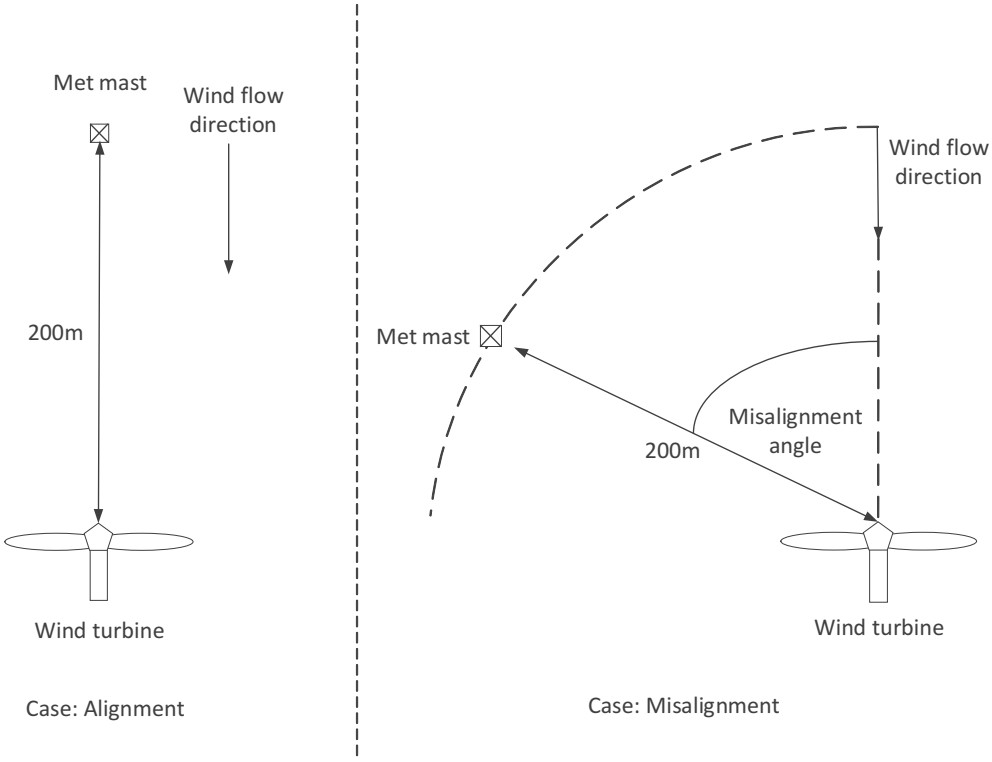

**Figure 4.** Effect of inflow angle on lateral offset of meteorological mast from wind turbine. Lateral offset is distance orthogonal to direction of wind flow.

In the case of misalignment, the spatial variance of the second-order moment of wind speed between mast and turbine increases with increasing misalignment. Figure 5 shows the effect of the misalignment angle on the normalized spatial variance of the second-order moment of wind speeds $u$, $v$ and $w$. The misalignment angle is defined as the angle between the wind direction and the line connecting mast and turbine. The results are obtained from simulations based on the Mann model. The simulation set-up and atmospheric conditions are the same as described in section 3.1.1.

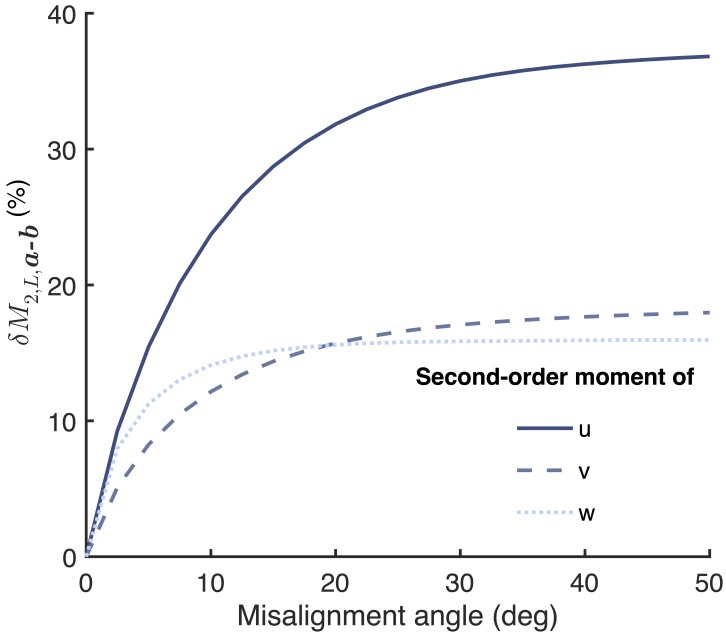

**Figure 5.** Effect of misalignment angle on normalized spatial variance of the second-order moment of wind speeds $u$, $v$ and $w$ averaged over 10min. Distance between wind turbine and meteorological mast is 200m, as described in Figure 4.

With misalignment, the flow measured at the mast is offset to the flow, that the wind turbine faces, in the cross-axial direction. As shown in Figure 1, such offset results in a spatial variance of the second-order moment of wind speed. As a result, the second-order moment of wind speed measured at the mast is associated with a random error compared to the second-order moment present at the wind turbine for reference.

It can be observed that the random error increases rapidly with increasing misalignment. The error reaches 90% of the asymptotic value at a misalignment of $22°$, $22°$, and $11°$ for the second-order moment of the wind velocity components $u$, $v$ and $z$, respectively. The asymptotic value of the error is 36%, 18% and 16% for the second-order moment of the wind velocity components $u$, $v$ and $z$, respectively.

It is therefore of interest to investigate the impact of such random error on the uncertainty of the measured fatigue loads and power output of the wind turbine. In Eggers et al. (2003) it is reported that a 70% increase in turbulence intensity resulted in an approximately tenfold increase in the fatigue damage fraction of the flapwise blade-root bending moment. In Saranyasoontorn

and Manuel (2008) a variance of turbulence intensity of 22.8% resulted in a variance of the damage-equivalent load (DEL) of the yaw moment of 12.7%. It is therefore concluded that uncertainty in the measured turbulence intensity can have a significant impact on the uncertainty in the measured fatigue loads. The impact of turbulence intensity on the power curve depends on the operational region of the wind turbine. In Clifton and Wagner (2014) it can be observed that the sensitivity of the power curve to turbulence intensity is small when the turbine is operating below the rated rotational speed. In operation at the rated rotational speed, the sensitivity is larger.

To mitigate the effect of the random error in the measured second-order moment on the classification of wind turbine performance, we recommend use of either or both of the following methods. First, the measurements of wind turbine performance can be filtered to only contain data for small angles of misalignment between wind direction and the line connecting mast and wind turbine. As shown in Figure 5, the error increases rapidly with misalignment. Thus, to limit the error to, for example, below 15%, the misalignment angle could be filtered for the range of $\pm 5°$.

In the second approach, the random error is mitigated by averaging turbine performance over ensembles of the same wind conditions. As such, the measured turbine performance is classified according to mean wind speed, turbulence intensity and atmospheric stability. In case turbine performance is wind direction-dependent, due to, for instance, topographical effects, the classification also needs to be performed with respect to wind direction. As a result, in each bin the wind conditions can be assumed to be on average the same at the meteorological mast and the wind turbine. Thus, the average wind conditions at the mast can be related to the average performance of the wind turbine for each wind condition bin. Consequently, the spatial variance of the second-order moment of wind speed is mitigated successfully.

### 3.2.3   Spatially Separated Sensor Verification

As a result of physical and economic constraints, sensors for the measurement of the second-order moment of wind speed are often verified using spatially separated reference measurements. Due to the distance between the sensor and reference, the result of the verification can be corrupted by the spatial variance of the second-order moment of wind speed.

This phenomenon is discussed in the following on the example of the verification case in Mittelmeier et al. (2016). The study compares turbine-based measurements indicative of turbulence intensity with reference measurements at an adjacent meteorological mast. The measurements at the wind turbine are wind speed, obtained from the nacelle anemometry, and the turbine power output. Turbulence intensity is directly calculated from the wind speed measurements. In the case of turbine power, the ratio of the standard deviation of power to its mean is considered as a measure related to turbulence intensity. One of the investigated wind farms is the Nordsee Ost wind farm with layout of wind turbines and meteorological mast as shown in Figure 6. The comparison of turbulence intensity measurements is conducted between the meteorological mast and wind turbine NO47. The study reports a Pearson correlation coefficient of 0.68 between turbulence intensity measurements at the meteorological mast and the nacelle anemometry of the wind turbine. The observed low correlation coefficient originates from the spatial variance of the second-order moment of wind speed, as is shown in the following.

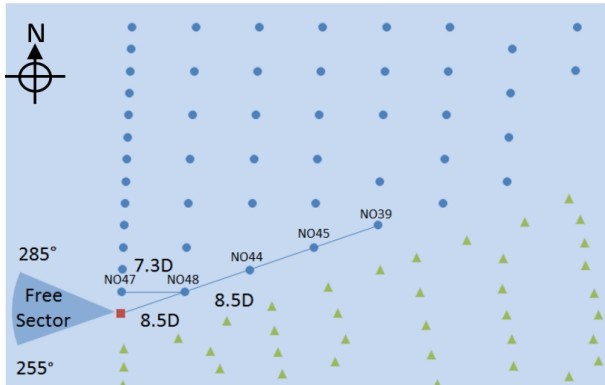

**Figure 6.** Nordsee Ost (blue cycles) with neighbouring wind farm Meerwind Süd (green triangles) and meteorological mast (red square) (Mittelmeier et al., 2016).

Assuming that in the investigated scales the largest spatial variance lies in the variability of wind speed and not its mean, the Pearson correlation coefficient of turbulence intensity $\rho_{TI,\boldsymbol{a},\boldsymbol{b}}$ can be approximated as

$$\rho_{TI,\boldsymbol{a},\boldsymbol{b}} = \frac{\mathrm{cov}\left(TI_{L,\boldsymbol{a}}(T), TI_{L,\boldsymbol{b}}(T)\right)}{\sigma^2(TI_L(T))} \approx \frac{\frac{1}{u_L^2}\mathrm{cov}\left(\sigma_{L,\boldsymbol{a}}(T), \sigma_{L,\boldsymbol{b}}(T)\right)}{\frac{1}{u_L^2}\sigma^2(\sigma_L(T))} = \frac{\mathrm{cov}\left(\sigma_{L,\boldsymbol{a}}(T), \sigma_{L,\boldsymbol{b}}(T)\right)}{\sigma^2(\sigma_L(T))} \tag{17}$$

where turbulence intensity $TI_{L,\boldsymbol{a}}(T)$ at a point $\boldsymbol{a}$ is defined as the ratio of the standard deviation of wind speed $\sigma_{L,\boldsymbol{a}}(T)$
5  to the mean wind speed $u_L$. Next, a linear relation of the second-order moment of wind speed and its square root is assumed in the proximity of its mean value. As a result, the Pearson correlation coefficient of the standard deviation of wind speed $\frac{\mathrm{cov}\left(\sigma_{L,\boldsymbol{a}}(T)\sigma_{L,\boldsymbol{b}}(T)\right)}{\sigma^2(\sigma_L(T))}$ can be approximated by the correlation coefficient of the second-order moment of wind speed as

$$\rho_{TI,\boldsymbol{a},\boldsymbol{b}} = \frac{\mathrm{cov}\left(\sigma_{L,\boldsymbol{a}}(T), \sigma_{L,\boldsymbol{b}}(T)\right)}{\sigma^2(\sigma_L(T))} \approx \frac{\mathrm{cov}\left(\mu_{2,L,\boldsymbol{a}}(T), \mu_{2,L,\boldsymbol{b}}(T)\right)}{\sigma^2(\mu_{2,L}(T))} \tag{18}$$

For a homogeneous turbulent field the covariance of the second-order moment of wind speed can be formulated as

$$\frac{\mathrm{cov}\left(\mu_{2,L,\boldsymbol{a}}(T), \mu_{2,L,\boldsymbol{b}}(T)\right)}{\sigma^2(\mu_{2,L}(T))} = \frac{\langle \mu_{2,L,\boldsymbol{a}}(T)\mu_{2,L,\boldsymbol{b}}(T)\rangle - \langle \mu_{2,L}(T)\rangle^2}{\sigma^2(\mu_{2,L}(T))} \tag{19}$$

Above Eq. 19 can be related to the spatial variance of the second-order moment of wind speed using Eq. 4 as

$$\rho_{TI,\boldsymbol{a},\boldsymbol{b}} \approx \frac{\langle \mu_{2,L,\boldsymbol{a}}(T)\mu_{2,L,\boldsymbol{b}}(T)\rangle - \langle \mu_{2,L}(T)\rangle^2}{\sigma^2(\mu_{2,L}(T))} = \frac{\langle \mu_{2,L}^2(T)\rangle - \frac{1}{2}\delta\mu_{2,L,\boldsymbol{a}-\boldsymbol{b}}^2(T) - \langle \mu_{2,L}(T)\rangle^2}{\sigma^2(\mu_{2,L}(T))} \tag{20}$$

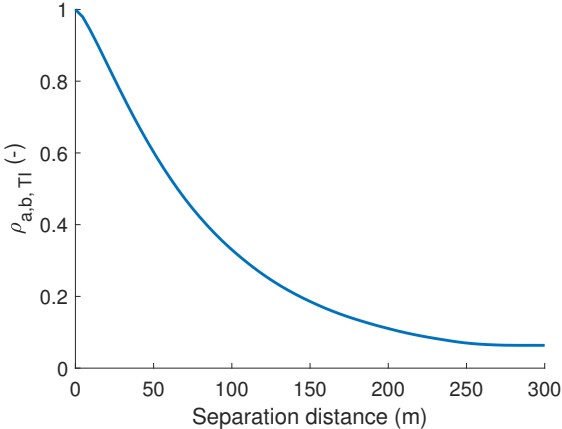

**Figure 7.** Impact of horizontal cross-flow separation distance between points $\boldsymbol{a}$ and $\boldsymbol{b}$ on correlation coefficient of turbulence intensity between these points.

Consequently, the Pearson correlation coefficient of turbulence intensity can be related to the spatial variance of the second-order moment of wind speed as

$$\rho_{TI,\boldsymbol{a},\boldsymbol{b}} \approx 1 - \frac{1}{2} \frac{\delta\mu_{2,L,\boldsymbol{a}-\boldsymbol{b}}^2(T)}{\sigma^2(\mu_{2,L}(T))} \tag{21}$$

Above Eq. 21 shows the impact of the spatial variance on the correlation coefficient. As such, larger spatial variance results
in a smaller correlation coefficient and vice versa. Due to the large spatial distance between measurement locations in the case
of the work of Mittelmeier et al. (2016), we can argue that a major contributor for the low correlation coefficient is the large
spatial variance of the second-order moment of wind speed.

To demonstrate this the correlation coefficient was quantified by calculation of the components of Eq. 21 from a simulated
turbulent field. The simulated field was generated as described in section 3.1.1. The simulated conditions are neutral atmo-
spheric boundary layer stability, a mean wind speed $U$ of 8.33m/s and a wind direction perpendicular to the line connecting
turbine and meteorological mast. Neutral atmospheric boundary layer stability is characterised using the Mann model with
parameters $\alpha\epsilon^{\frac{2}{3}} = 1$, $l = 50$m, $\Gamma = 3.2$, according to Sathe et al. (2013). The averaging duration $T$ is set to 600s.

Figure 7 shows the impact of the horizontal cross-flow separation distance between points $\boldsymbol{a}$ and $\boldsymbol{b}$ on the correlation co-
efficient of turbulence intensity between these points. It can be observed that the correlation coefficient declines rapidly with
increasing separation distance. For a distance greater than 200m it is smaller than 0.1. In the Nordsee Ost wind farm, the
cross-flow distance between mast and turbine is larger than 400m. Hence, a low correlation coefficient of similar magnitude
as in the simulations would be expected. The larger correlation coefficient observed in the measurements is likely because the

correlation coefficient was obtained from measurements of various conditions of atmospheric boundary layer stability. This can be understood from the definition of the correlation coefficient as follows.

$$\rho_{TI,\boldsymbol{a},\boldsymbol{b}} = \frac{\left\langle \left( TI_{L,\boldsymbol{a}}(T) - \langle TI_{L,\boldsymbol{a}}(T) \rangle \right) \left( TI_{L,\boldsymbol{b}}(T) - \langle TI_{L,\boldsymbol{b}}(T) \rangle \right) \right\rangle}{\sigma^2(TI_L(T))} \tag{22}$$

Turbulence intensity differs more between unstable and stable atmospheric boundary layer conditions than due to spatial separation of measurement locations. As a result, spatially separated measurements of turbulence intensity of the same atmospheric condition appear correlated with respect to the average turbulence intensity of all stability conditions. Consequently, the blending of measurements from different atmospheric conditions results in a larger correlation coefficient, as observed in the Nordsee Ost case.

Overall, the results demonstrate the low correlation of measurements of turbulent flow in case of separation in cross-flow distance. It is therefore of interest to develop more accurate approaches for the verification of sensors in case of spatially-separated reference measurements. Our recommended approach is to filter for direction misalignment, that is the angle between the wind direction and the line connecting sensor and reference, or to average over ensembles of the same atmospheric conditions. The approach is equivalent to the mitigation measures discussed with regards to the verification of wind turbine performance, described in section 3.2.2. The interested reader is thus referred to that section for more details on the approach.

## 4    Conclusions

The first analytical solution for the quantification of the spatial variance of second-order moment of wind speed was developed in this work. The spatial variance is defined as random differences in the sample variance of wind speed between different points in space. The approach is successfully verified using simulation and field data. The impact of the spatial variance of the second-order moment of wind speed is then investigated in three, selected applications of the wind energy sector including mitigation measures. First, the variance of the second-order moment between front-row wind turbines of Lillgrund wind farm is investigated. The variance ranges between 25% and 48% for turbulence intensities ranging from 7% to 10%. Using the average turbulence intensity at front-row turbines as estimate for ambient turbulence intensity would thus result in a random error in flow model inputs. It is thus suggested to use the second-order moment measured at each individual turbine as input to flow models in order to mitigate the random error. This is particularly of importance for dynamic flow models used for wind farm control as these aim to capture the dynamics of flow rather than average properties. Second, the impact of the spatial variance of the measured second-order moment on the verification of wind turbine performance is investigated. Misalignment between the mean wind direction and the line connecting the meteorological mast and wind turbine results in a random error in the observed second-order moment. Such random error results in uncertainty in the turbulence intensity-based classification of the fatigue loads and power output of the wind turbine. To mitigate the random error, it is suggested to either filter the measured data for low angles of misalignment or to quantify wind turbine performance using the ensemble average over the same wind conditions. Third, the verification of sensors in wind farms can involve distant reference measurements. In case

of a misalignment between the wind direction and the line connecting sensor and reference, a random error will hamper the comparison of second-order moments measured at distant locations. Similar to the verification of turbine performance, filtering the measured data for low angles of misalignment or using the ensemble average, can mitigate the random error.

To conclude, the comparison or combination of measurements of the second-order moment of wind speed from spatially separated locations can result in a random error. Assuming Taylor's hypothesis of frozen turbulence, the random error is particularly prominent for the separation in the cross-axial and vertical direction of measurement locations. This work shows that knowledge of the drivers of the random error allows for mitigation measures.

*Competing interests.* There is no competing interests.

*Acknowledgements.* This work is partially funded by the CONCERT project, which is financed by Energinet.dk under the Public Service Obligation scheme (ForskEL 12396). We also thank the Danish Energy Agency for funding through the New European Wind Atlas project (EUDP 14-II).

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
