# Peer review of "Mitigating Impact of Spatial Variance of Turbulence in Wind Energy Applications"

_Wind Energy Science, 2019_

## Referee Comment (RC1) · Anonymous Referee #1 · 24 Apr 2019

The reviewed manuscript by Jonas Kazda and Jakob Mann presents, as the authors emphasize, the first analytical solution for the quantification of the spatial variance of the second-order moment of correlated wind speeds – which is a very valuable, and well presented, contribution. The authors further introduce three examples of applications and describe for these how the analytically quantified impact can be mitigated. This second part is in my view rather weak at the current stage and should be further elaborated. Apart from that the three individual examples are treated in different detail, the mitigation approaches are not very convincing from a practical view. This point of criticism will be further detailed in the following comments.

Specific comments (following the order of the manuscript):

- Paragraph starting page 2, line 24 / general comment to application "2" – In power

curve testing according to the current standards, the turbulence influence is rather considered a mathematical issue. The actual influence of turbulence on the performance is not yet accounted for. So, I am not sure how relevant this example is here.

- Paragraph starting p. 3 l. 1 – For application "3" it is not clear to me what kind of "sensors" are referred to. Maybe the example is presented in a too general and sufficiently clear way here.

- p. 6 l. 16 – The authors state that Figure 1 shows an "overall agreement". This conclusion should be justified in a better way. Is the agreement really good enough? (what is the reference level?)

- p. 11 / conclusion of 3.2.1 – As I understand the description, the analytical model is here only used to explain the observed variations, making a location-specific TI measurement necessary. Is this really a mitigation? I am wondering if the model could not also be used for a correction. (This would be my understanding of a more practical mitigation. . .)

- 3.2.3 is really short with not a lot of content. The authors should either further elaborate on this section or possibly leave it out. I could imagine that the approach could here be used for a detailed uncertainty assessment, again as a more practical outcome. Technical corrections: (Only a selection here, a more detailed language review is recommended.)

- Page 1, line 1 – I would suggest a rewording of the first words. . . something like "For the first time. . ."

- p. 2 l. 30 – I believe what the authors mean is the uncertainty of the power curve, and not the power itself.

- p. 3 l. 11 (and several other places in the document) – Comma signs in "expected, spatial" and "two, spatially-" are unnecessary.

- (6) and p. 4 l. 10 – The equation needs to be revised. There is neither a vector r nor

a delta t.

- Figure 1 – I assume the y-axis corresponds to \delta M – would be a more handy label.

---

## Referee Comment (RC2) · Anonymous Referee #2 · 4 Jul 2019

This is certainly an interesting paper, with a good introduction, and interesting outcomes. It makes a number of assumptions about the turbulent flow approaching the front row of turbines, and takes data from Lillgrund to verify the results of the analysis. It is obvious that for a short averaging period (T) the turbulence approaching one turbine will not have the same statistics as that approaching another, even for turbulence that is laterally homogeneous when averaged over a long time. But it is worth making the point too. It would be nice to know how the predictions vary with T. It's also obvious, is it not?, that the nacelle instrumentation (assuming it is working) should give a better indication than remote mast instrumentation – a point of conclusion on page 11, line 18.

Through to equation 4 is straightforward, but I cannot comment adequately on the

remainder, bar a few points. On page 4, line 2, the mean velocity is put to zero. This seems rather odd (i.e. wrong), and so needs some justification. Line 10, the vector r is not defined.

Page 5 sentence on line 16 is essentially a repeat of that on line 14. The Mann model is for neutral flow, so something should be said when applying the results to supposedly (weakly) non-neutral flow. Suggest a bit more is said about the bands in Fig 3. Cite ref.

Page 7, line 15. Presume this should say eqn 9.

Fig 3. In comparing this with Fig 2 it would be easier of the left-right order was reversed - to be the same as Fig 2.

Page 9, line 4. "A similar, spatial .." . This seems to be repeating what's covered in the previous chapter?

Page 11, line 2. The frequency range in the near wake is bound to be to much higher frequencies than in the upstream flow because the blade-wake scale, as they form into the near wake, are much smaller, and also energetic.

The assumption of frozen flow (Taylor's hypothesis) – it's validity, or lack of - will have offsetting influences: if the wind is stable the ABL will be less deep (wrt neutral), and so a fixed distance –eg 200m – will be a greater relative distance – making the assumption less valid. On the other hand, the lower turbulence intensity will have the opposite effect. The opposite occurs for unstable flow: deeper, so relatively closer vs more intense.

Page 13, line 14. "The second approach ..." First approach is fine, it's clear, but the second is not clear/obvious. I think it needs some elaboration.

Overall, I think the latter half of the paper is not as well written as it could be.

---

## Author Comment (AC1) · 20 Aug 2019

Dear Referee,

Please find our response attached.

Please also note the supplement to this comment:
https://www.wind-energ-sci-discuss.net/wes-2019-10/wes-2019-10-AC1-supplement.pdf

---

## Author Comment (AC2) · 20 Aug 2019

Dear Referee,

Please find our response attached.

Please also note the supplement to this comment:
https://www.wind-energ-sci-discuss.net/wes-2019-10/wes-2019-10-AC2-supplement.pdf

---

## Author Response (AR1)

**Referee #1**

The reviewed manuscript by Jonas Kazda and Jakob Mann presents, as the authors emphasize, the first analytical solution for the quantification of the spatial variance of the second-order moment of correlated wind speeds – which is a very valuable, and well presented, contribution. The authors further introduce three examples of applications and describe for these how the analytically quantified impact can be mitigated. This second part is in my view rather weak at the current stage and should be further elaborated. Apart from that the three individual examples are treated in different detail, the mitigation approaches are not very convincing from a practical view. This point of criticism will be further detailed in the following comments.

**Authors' response:**

We would like to thank the reviewer for the comments and time spent with reviewing the manuscript. All comments are addressed and discussed below. Along with the comments of the reviewer the application areas were strengthened and extended, as suggested by the reviewer. Please find the revised manuscript with changes highlighted in yellow after the response.

Paragraph starting page 2, line 24 / general comment to application "2" – In power curve testing according to the current standards, the turbulence influence is rather considered a mathematical issue. The actual influence of turbulence on the performance is not yet accounted for. So, I am not sure how relevant this example is here.

**Authors' response:**

According to (Clifton and Wagner, 2014), "a revision of the IEC standard is in progress and the new edition addresses vertical shear, directional shear and turbulence". We are therefore convinced of the relevance of the topic. To reflect this development, we have extended the statement on page 2, line 24 as follows.

*"Next, it is increasingly common to use turbulence intensity measurements for the classification of turbine performance. Having been discussed in literature over the past decade, the impact of turbulence intensity on turbine performance is addressed in the next revision of the IEC standard (IEC, 2005)."*

Paragraph starting p. 3 l. 1 – For application "3" it is not clear to me what kind of "sensors" are referred to. Maybe the example is presented in a too general and sufficiently clear way here.

**Authors' response:**

We thank the reviewer for the comment. We have extended the discussion of application '3' in the introduction and results as follows.

*"The third application area discussed in the present work is the spatially separated verification of sensors for the measurement of the second-order moment of wind speed. In such case, the measurements of the to-be-verified sensor are compared to a spatially distant reference measurement. Due to the distance between the sensor and reference, the result of the verification can be corrupted by the spatial variance of the second-order moment of wind speed. This phenomenon is discussed in the present work on the example of the verification case in Mittelmeier et al. (2016). Here turbine-based measurements of turbulence intensity are verified with reference measurements at an adjacent meteorological mast."*

*"As a result of physical and economic constraints, sensors for the measurement of the second-order moment of wind speed are often verified using spatially separated reference measurements. Due to the distance between the sensor and reference, the result of the verification can be corrupted by the spatial variance of the second-order moment of wind speed.*

*This phenomenon is discussed in the following on the example of the verification case in Mittelmeier et al. (2016). The study compares turbine-based measurements indicative of turbulence intensity with reference measurements at an adjacent meteorological mast. The measurements at the wind turbine are wind speed, obtained from the nacelle anemometry, and the turbine power output. Turbulence intensity is directly calculated from the wind speed measurements. In the case of turbine power, the ratio of the standard deviation of power to its mean is considered as a measure related to turbulence intensity. One of the investigated wind farms is the Nordsee Ost wind farm with layout of wind turbines and meteorological mast as shown in Figure 6. The comparison of turbulence intensity measurements is conducted between the meteorological mast and wind turbine NO47. The study reports a Pearson correlation coefficient of 0.68 between turbulence intensity measurements at the meteorological mast and the nacelle anemometry of the wind turbine. The observed low correlation coefficient originates from the spatial variance of the second-order moment of wind speed, as is shown in the following.*

… See further details in manuscript …

*Overall, the results demonstrate the low correlation of measurements of turbulent flow in case of separation in cross-flow distance. It is therefore of interest to develop more accurate approaches for the verification of sensors in case of spatially-separated reference measurements. Our recommended approach is to filter for direction misalignment, that is the angle between the wind direction and the line connecting sensor and reference, or to average over ensembles of the same atmospheric conditions. The approach is equivalent to the mitigation measures discussed with regards to the verification of wind turbine performance, described in section 3.2.2. The interested reader is thus referred to that section for more details on the approach."*

p. 6 l. 16 – The authors state that Figure 1 shows an "overall agreement". This conclusion should be justified in a better way. Is the agreement really good enough? (what is the reference level?)

**Authors' response:**

We are convinced that the agreement is sufficient. The reasons for our conclusion are thoroughly discussed in the paragraph ranging from page 6, line 16 to page 7, line 8.

p. 11 / conclusion of 3.2.1 – As I understand the description, the analytical model is here only used to explain the observed variations, making a location-specific TI measurement necessary. Is this really a mitigation? I am wondering if the model could not also be used for a correction. (This would be my understanding of a more practical mitigation...)

**Authors' response:**

The analytical solution and measurements are both used to quantify the spatial variance of the second-order moment of wind speed between wind turbines in section 3.2.1. On page 9, line 2 the spatial variance is quantified using the analytical solution. The quantification based on measurements is discussed along with Figure 3. The measurement results are used to verify the analytical solution and to exemplify the magnitude of the spatial variance of the second-order moment of wind speed in a real wind farm.

The mitigation of the spatial variance is discussed in section 3.2.1 with respect to the use of turbulence intensity measurements in dynamic flow models. The results show an up to 48% spatial variance of the second-order moment of wind speed between turbines. It is therefore concluded that the turbulence intensity measured at each turbine individually shall be used as input to dynamic flow models; in order to mitigate the impact of the spatial variance of turbulence.

Since the spatial variance of the second-order moment of wind speed is a random error, it is unclear to the authors how the reviewer suggests to corrected it.

3.2.3 is really short with not a lot of content.  The authors should either further elaborate on this section or possibly leave it out.  I could imagine that the approach could here be used for a detailed uncertainty assessment,  again as a more practical outcome. Technical corrections: (Only a selection here, a more detailed language review is recommended.)

**Authors' response:**

Along with an earlier comment of the reviewer, Section 3.2.3 was extended to include a more detailed discussion of the topic.

Page 1, line 1 – I would suggest a rewording of the first words...something like "For the first time..."

**Authors' response:**

We thank the reviewer for the comment. The sentence was updated as suggested and now reads as follows.

*"For the first time an analytical solution for the quantification of the spatial variance of the second-order moment of correlated wind speeds was developed in this work."*

p. 2 l. 30 – I believe what the authors mean is the uncertainty of the power curve, and not the power itself.

**Authors' response:**

We thank the reviewer for the comment. The sentence was updated as follows.

*"Uncertainty in the measured turbulence intensity propagates into the uncertainty of the measured power curve and fatigue loads of the wind turbine."*

p. 3 l. 11 (and several other places in the document) – Comma signs in "expected, spatial" and "two, spatially-" are unnecessary.

**Authors' response:**

The revised manuscript was reviewed for the use of comma signs.

(6) and p. 4 l. 10 – The equation needs to be revised. There is neither a vector r nor a delta t.

**Authors' response:**

The definition of vector *r* and *delta t* was moved to page 4, line 9, so it occurs immediately after its first use.

Figure 1 – I assume the y-axis corresponds to\delta M – would be a more handy label.

**Authors' response:**

We thank the reviewer for the comment. The label of the y-axis was updated as suggested.

**Referee #2**

This is certainly an interesting paper, with a good introduction, and interesting outcomes.

**Authors' response:**

We would like to thank the reviewer for the comments and time spent with reviewing the manuscript. We are grateful for the appreciation of the work. All comments are addressed and discussed below. Please find the revised manuscript with changes highlighted in yellow after the response.

It makes a number of assumptions about the turbulent flow approaching the front row of turbines, and takes data from Lillgrund to verify the results of the analysis. It is obvious that for a short averaging period (T) the turbulence approaching one turbine will not have the same statistics as that approaching another, even for turbulence that is laterally homogeneous when averaged over a long time. But it is worth making the point too. It would be nice to know how the predictions vary with T. It's also obvious, is it not?, that the nacelle instrumentation (assuming it is working) should give a better indication than remote mast instrumentation – a point of conclusion on page 11, line 18.

**Authors' response:**

The conclusion that the second-order moment of wind speed varies between adjacent turbines is new to literature. As such, typically, a uniform ambient turbulence intensity is assumed, such as in (Niayifar, A. and Porté-Agel, F., 2016), for instance. The authors are therefore convinced that the spatial variance of the second-order moment of wind speed is not obvious to the research community of wind farm flow modelling and control.

The conclusion on page 11, line 18 does not compare nacelle instrumentation with a met mast. It states that in dynamic wind farm flow models the use of locally measured turbulence intensity at wind turbines is more accurate than averaging over turbulence intensity measurements at upstream turbines.

Through to equation 4 is straightforward, but I cannot comment adequately on the remainder, bar a few points. On page 4, line 2, the mean velocity is put to zero. This seems rather odd (i.e. wrong), and so needs some justification. Line 10, the vector r is not defined.

**Authors' response:**

We thank the reviewer for the comment. Indeed, the mean wind speed is typically larger than zero in wind energy applications. The assumption of a zero mean wind speed is not necessary since assuming a non-zero mean wind speed gives the exact same results, but it makes the equations much more compact. To clarify this to the reader, we have extended the assumption as follows.

*"Next, it is assumed that the mean wind speed $u_L$ is zero. This assumption is not necessary since assuming a non-zero mean wind speed gives the exact same results, but it makes the equations much more compact."*

The definition of the vector *r* is extended to the following.

*"… where **r** is a three-dimensional vector connecting the two points …"*

Page 5 sentence on line 16 is essentially a repeat of that on line 14. The Mann model is for neutral flow, so something should be said when applying the results to supposedly (weakly) non-neutral flow. Suggest a bit more is said about the bands in Fig 3. Cite ref.

**Authors' response:**

We thank the reviewer for the comment. We have adapted line 14 as follows.

*"The analytical solution (Eq. 9) is successfully verified in the following against a simulated wind field."*

Albeit the Mann model is derived for neutral flow, the analytical solution of the spatial variance of the second-order moment of wind speed is applicable to any atmospheric stability regime.

The discussion of the bands in Fig. 3 spans on page 11 from line 7 to line 16 and was elevated to a more general level as follows.

*"In general, when moving across the stability spectrum from stable to unstable the integral length scale increases (Sathe et al. 2013). As a result of this increase, the asymptotic value of the spatial variance of the second-order moment of wind speed increases, according to Eq. 14."*

Page 7, line 15. Presume this should say eqn 9.

**Authors' response:**

We thank the reviewer for the comment. This is our oversight.

Fig 3. In comparing this with Fig 2 it would be easier of the left-right order was reversed- to be the same as Fig 2.

**Authors' response:**

Fig. 2 was updated to the same order as Fig. 3.

Page 9, line 4. "A similar, spatial .." . This seems to be repeating what's covered in the previous chapter?

**Authors' response:**

We do not see a repetition with the previous chapter, i.e. 3.1. In case the reviewer refers to the previous paragraph, we added the following statement to distinguish the analysis of these more clearly.

*"In the following, the spatial variance is first investigated using the analytical solution, and thereafter using measurements."*

Page 11, line 2. The frequency range in the near wake is bound to be to much higher frequencies than in the upstream flow because the blade-wake scale, as they form into the near wake, are much smaller, and also energetic.

**Authors' response:**

We agree with the reviewer. This is already described on page 10, line 16 onwards.

The assumption of frozen flow (Taylor's hypothesis) – it's validity, or lack of - will have offsetting influences: if the wind is stable the ABL will be less deep (wrt neutral), and so a fixed distance –eg 200m – will be a greater relative distance – making the assumption less valid. On the other hand, the lower turbulence intensity will have the opposite effect. The opposite occurs for unstable flow: deeper, so relatively closer vs more intense.

**Authors' response:**

The reviewer is right in pointing out that failure of Taylor's hypothesis would change the results. However, what the reviewer describes is how the turbulence length scale changes with stability, which is another issue, which we address by referencing to the Sathe et al paper. Failure of Taylor's hypothesis would mostly affect situations where one measurement position is directly downstream of the other. In this situation our model states that except from a short period in the start and the end of the measurement period the time series measured would be identical. In reality this is of course not the case and it would lead to that our theory would under predict the difference in the measured variances. We have added the following statement to the introduction and results in Section 3.2.2.

*"The assumption may lead to an underestimation of the difference in variances in the situation where one measurement position is more or less directly downstream from the other."*

*"In the case of alignment, the random error due to the spatial variance of turbulence can be regarded as small assuming Taylor's hypothesis of frozen turbulence."*

We feel that further discussion of this is not warranted since it is difficult to assess this bias.

Page 13, line 14. "The second approach ..." First approach is fine, it's clear, but the second is not clear/obvious. I think it needs some elaboration. Overall, I think the latter half of the paper is not as well written as it could be.

**Authors' response:**

We have rephrased the second approach as follows to improve understanding and clarity.

[revised manuscript text omitted]

---

## Referee Report (RR1)

Mitigating Impact of Spatial Variance of Turbulence in Wind Energy Applications, by Kasda and Mann Wind Energy Science. Revised version

The following are points the authors could consider.

Page 2, line 18. "There are", rather than "There is". Line 25. "performance will be addressed"

Page 3, line 3. "variation of spatially separated sensors" Line 5. "between sensor and reference" or "between the sensor and the reference"

Page 6, line 15. "The remaining ... remained infinite." What *does* this mean? And 'infinite' – is this something to worry about?

Page 7. Would a physical argument also help? "Physically, as the points move sufficiently far apart the variance must cease to change, simply because the turbulence is no-longer correlated. It is the fact that the turbulence is correlated (over a distance less than order the integral length scale) that the variance will be reduced below an asymptotic level, necessarily falling to zero at zero separation. The form of Fig 1 is therefore as expected." Qualitative physical arguments are worth having.

Also, could a boundary layer scale -e.g. an overall height -be given in association with Fig 1 and Table 1?

And it would still be interesting to know how the asymptotic levels change with T. Could some example be given? – of halving or doubling?

Page 8, line 27. "Thereby the local realizations of turbulent structures can be taken into account." Is this really the case? There is not much structural information, is there?

Page 9, line 1. Suggest "between turbines arises from the spatial". Line 2. ", an investigation was made on the".

Case of Fig 2. Presume ABL conditions of Fig 1, Table 1 are typical of those relevant here. Should this be said? (Some overall ABL height would be useful.)

Page 10+11. What level of turbulence intensity is taken to be indicative of neutral conditions? 7%?

Page 11. Line 22 and line 24. 'more unstable conditions' Repetition. This needs tidying up.

Fig 4, right hand fig. Could not the misalignment angle be indicated by a circular arc, and so look nicer?

Figs 3 and 5. Should not the vertical axis have same label as Fig 1?

Page 14. Paragraph from line 8. Should not wind direction appear explicitly in the list?

Page 16, line 14. I don't understand this paragraph.

---

## Author Response (AR2)

Dear Jonas Kazda,

We are pleased to inform you that the Associate Editor report for the following manuscript is now available:

Journal: WES
Title: Mitigating Impact of Spatial Variance of Turbulence in Wind Energy Applications
Author(s): Jonas Kazda and Jakob Mann
MS No.: wes-2019-10
MS Type: Research article
Iteration: Minor Revision

The Associate Editor has decided that minor revisions are necessary before the manuscript can be accepted. Please find the Associate Editor Report at https://editor.copernicus.org/WES/ms_records/wes-2019-10.

We kindly ask you to revise your manuscript accordingly and to upload the revised files, a point-by-point reply to the comments, and a marked-up manuscript version showing the changes made in your File Manager no later than 06 Dec 2019: https://editor.copernicus.org/WES/file_manager/wes-2019-10. Please find all information on manuscript submission under https://www.wind-energy-science.net/for_authors/submit_your_manuscript.html.

Your revised manuscript will be reviewed by the Associate Editor and you will be informed about the outcome by separate email.

Besides adjustments requested by the Associate Editor or Referees, please check your manuscript carefully for typos, missing co-authors and their affiliations, terminology, updates of data in tables, or updates of variables in equations. All these have to be clarified with the Associate Editor and therefore have to be included before you submit your revised manuscript. Should your manuscript be finally accepted it will not be possible to include such rather substantial changes anymore when your manuscript is in final production (proofreading).

To log in, please use your Copernicus Office user ID 457063.

Please note that all Referee and Associate Editor reports, the author's response, as well as the different manuscript versions of the peer-review completion (post-discussion review of revised submission) will be published if your paper will be accepted for final publication in WES.

You are invited to monitor the processing of your manuscript via your MS Overview: https://editor.copernicus.org/WES/my_manuscript_overview

In case any questions arise, please contact me. Thank you very much for your cooperation.

Kind regards,

Natascha Töpfer
Copernicus Publications
Editorial Support
editorial@copernicus.org

on behalf of the WES Editorial Board

**Authors' response:**
We are very grateful for the associate editors report, and for the reviewers' comments and their time spent with the review. All comments are addressed and discussed below.

**Anonymous Referee #2**

The following are points the authors could consider.

**Authors' response:**

We kindly thank the reviewer for the time he / she spent with the review of the manuscript. Please find our answers to your comments below.

Page 2, line 18. "There are", rather than "There is". Line 25. "performance will be addressed". Page 3, line 3. "variation of spatially separated sensors". Line 5. "between sensor and reference" or "between the sensor and the reference"

**Authors' response:**

The phrases were updated as suggested by the reviewer.

Page 6, line 15. "The remaining ... remained infinite." What *does* this mean? And 'infinite' – is this something to worry about?

**Authors' response:**

The statement was updated to the following.

> *"The integration range of the other integrals of the analytical solution remained infinite."*

The value of the integral with infinite integration range is well defined for the investigated integrand in the present case.

Page 7. Would a physical argument also help? "Physically, as the points move sufficiently far apart the variance must cease to change, simply because the turbulence is no-longer correlated. It is the fact that the turbulence is correlated (over a distance less than order the integral length scale) that the variance will be reduced below an asymptotic level, necessarily falling to zero at zero separation. The form of Fig 1 is therefore as expected." Qualitative physical arguments are worth having.

**Authors' response:**

We have extended the physical argument along the comment of the reviewer as follows.

> *"At large separation distances, typically in the order of the integral length scale l, the second-order moment of wind speed at the points a and b becomes uncorrelated."*

Also, could a boundary layer scale – e.g. an overall height – be given in association with Fig 1 and Table 1?

**Authors' response:**

The boundary layer height does not directly enter into the Mann model, only indirectly through the length scale parameter. The length scale parameter is taken from the paper by Sathe et al (2013) and are believed to be representative of the neutral offshore boundary layer.

And it would still be interesting to know how the asymptotic levels change with T. Could some example be given? – of halving or doubling?

**Authors' response:**

The impact of the measurement duration T on the asymptotic level of the normalized spatial variance of the second-order moment of wind speed can be understood from Eq. 14. As such, for instance, halving the duration T will result in an increase of the asymptotic level by the square root of two.

Page 8, line 27. "Thereby the local realizations of turbulent structures can be taken into account." Is this really the case? There is not much structural information, is there?

**Authors' response:**

The statement was rephrased to the following.

*"Thereby the local realization of turbulent flow can be taken into account."*

Page 9, line 1. Suggest "between turbines arises from the spatial". Line 2. ", an investigation was made on the".

**Authors' response:**

The phrases were updated as suggested by the reviewer.

Case of Fig 2. Presume ABL conditions of Fig 1, Table 1 are typical of those relevant here. Should this be said? (Some overall ABL height would be useful.)

**Authors' response:**

The ABL conditions in the Lillgrund study are classified implicitly using turbulence intensity, as shown in Figure 2 and 3. The impact of ABL conditions on the comparison of the observations from the Lillgrund study with the analytical solution from Figure 1 is discussed on page 10, line 12 to page 11 line 27.

Page 10+11. What level of turbulence intensity is taken to be indicative of neutral conditions? 7%?

**Authors' response:**

Based on the observations in Figure 3, the measured turbulence intensity of 7% to 8% could indicate neutral conditions, as discussed on page 10 line 13 ff. Given that the turbulence intensity is measured at the turbines the ambient turbulence intensity is likely to be lower at neutral conditions.

Page 11. Line 22 and line 24. 'more unstable conditions' Repetition. This needs tidying up.

**Authors' response:**

The phrase was adjusted to circumvent word repetition. The statement now reads as follows.

> *"First, according to Sathe et al. (2013) the integral length scale is larger in unstable conditions than in stable ones, and based on Eq. 14 this results in a larger value of the asymptote."*

Fig 4, right hand fig. Could not the misalignment angle be indicated by a circular arc, and so look nicer?

**Authors' response:**

We believe the design of Figure 4 is clear and aesthetic.

Figs 3 and 5. Should not the vertical axis have same label as Fig 1?

**Authors' response:**

We thank the reviewer for the comment. We are convinced that the y-axis label of Figure 3 is sufficiently clear given the definitions provided in the paper. To increase clarity of Figure 5, the y-axis label was updated as suggested by the reviewer.

Page 16, line 14. I don't understand this paragraph.

**Authors' response:**

The explanation to page 16 line 14 is given in the subsequent paragraph.

Page 14. Paragraph from line 8. Should not wind direction appear explicitly in the list?

**Authors' response:**

Wind direction becomes relevant in case turbine performance is wind direction dependent due to, for example, topographical effects. The following statement was added to the discussion to clarify this.

> *"In case turbine performance is wind direction-dependent, due to, for instance, topographical effects, the classification also needs to be performed with respect to wind direction."*

**Anonymous Referee #3**

Very interesting paper, which I recommend for publication after improving some minor aspects.

**Authors' response:**

We would very much like to thank the reviewer for the time he / she spent with the review of the manuscript. Please find our answers to your comments below.

A citation for further information on Lillgrund wind farm should be provided.

**Authors' response:**

A selected citation was added to Section 3.2.1 on page 9 line 4 to provide further information on Lillgrund wind farm.

Fig 3 (and 5?) would be nice to write for the y-Axsis that this is again \delta M(…)

**Authors' response:**

We thank the reviewer for the comment. We are convinced that the y-axis label of Figure 3 is sufficiently clear given the definitions provided in the paper. To increase clarity of Figure 5, the y-axis label was updated as suggested by the reviewer.

There is a principal aspect of presentation which should be improved: The notation introduced in Chapter 2 should be taken as basis of the whole paper. Later on there are several quantities (related to the quantities of Chart 2.) which are not well introduced - mainly clear definitions are missing.

** for RMS of Fig 5 (and corresponding text)

** for TI , \rho and \sigma close to equ. 17

** The Pearson correlation - was already nearly used in equation 4.

**Authors' response:**

We thank the reviewer for the comment. The discussion of Figure 5 has been extended to provide an improved and more coherent understanding to the reader. Along with Eq. 17, the following definition was added on turbulence intensity TI and $\sigma$.

*"... where turbulence intensity $TI_{L,a}(T)$ at a point **a** is defined as the ratio of the standard deviation of wind speed $\sigma_{L,a}(T)$ to the mean wind speed $u_L$."*

$\rho_{TI,a,b}$ is the Pearson correlation coefficient defined above Eq. 17 in line 2 on page 15.

The Pearson correlation coefficient in Eq. 17 and the spatial variance of the second-order moment in Eq. 4 are similar, nonetheless different. This can be seen from the derivations of Eq. 17 – 20, which are required to relate these two quantities.

The results and presentations of Chapter 3.2.3 and Chapter 2 should be revised in the way that a common notation is used.

**Authors' response:**

The notations in Chapter 2 and Chapter 3.2.3 are coherent, as quantities are expressed using the same notation. This can be observed, for instance, in Eq. 18.

A simple explanation of the limits of \detla M for totally correlated and totally uncorrelated should be given. (Here I run in some problems with the results, if the signals at a und b are uncorrelated I expect that \delta M become sqrt{2} (see equ. 10) How can you explain the saturation of Fig 1 for 35%??.) I suggest that the authors discuss this in combination with equations. 21. also \rho should be related to \delta M.

**Authors' response:**

The asymptotic value of the normalized spatial variance $\delta M_{2,L,\infty}$ depends on the atmospheric conditions. As can be seen from Eq. 13, $\delta M_{2,L,\infty}$ does usually not yield the square root of two, since the standard deviation of the second-order moment $\sigma(\mu_{2,L}(T))$ typically differs from the expected second-order moment of wind speed $\langle\mu_{2,L}(T)\rangle$.

Are equations 14 and 15 is valid for \tau <= T and const for tau> T? in these equations I expected sort (2) and not 2

**Authors' response:**

The format of Eq. 14 and 15 is correct and is valid for $\tau \ll T$. The latter was added to the manuscript as follows.

*"This approximation is valid for $\tau \ll T$ and $L \ll l$."*

My suggestion to revise the paper with respect to the used symbols and formulas. Bring all in a consistent way. I have the feeling this can be done best in Chap2 (or you may use an appendix for the formal relations)

**Authors' response:**

As suggested by the reviewer, the use of symbols and formulas was align within the paper.